# β-RA Targets Mitochondrial Metabolism and Adipogenesis, Leading to Therapeutic Benefits against CoQ Deficiency and Age-Related Overweight

**DOI:** 10.3390/biomedicines9101457

**Published:** 2021-10-13

**Authors:** Agustín Hidalgo-Gutiérrez, Eliana Barriocanal-Casado, María Elena Díaz-Casado, Pilar González-García, Riccardo Zenezini Chiozzi, Darío Acuña-Castroviejo, Luis Carlos López

**Affiliations:** 1Departamento de Fisiología, Facultad de Medicina, Universidad de Granada, 18016 Granada, Spain; ahg@ugr.es (A.H.-G.); elianabc@ugr.es (E.B.-C.); elenadiaz@ugr.es (M.E.D.-C.); pgonzalez@ugr.es (P.G.-G.); dacuna@ugr.es (D.A.-C.); 2Centro de Investigación Biomédica, Instituto de Biotecnología, Universidad de Granada, 18016 Granada, Spain; 3Biomolecular Mass Spectrometry and Proteomics, Bijvoet Center for Biomolecular Research, Utrecht Institute for Pharmaceutical Sciences, University of Utrecht, Padualaan 8, 3584 CH Utrecht, The Netherlands; r.zenezinichiozzi@uu.nl; 4Netherlands Proteomics Centre, Padualaan 8, 3584 CH Utrecht, The Netherlands; 5Centro de Investigación Biomédica en Red de Fragilidad y Envejecimiento Saludable (CIBERFES), 18016 Granada, Spain

**Keywords:** mitochondrial disease, encephalopathy, astrogliosis, spongiosis, obesity, white adipose tissue, mitochondrial proteome, 3T3-L1, mouse model, hepatic steatosis

## Abstract

Primary mitochondrial diseases are caused by mutations in mitochondrial or nuclear genes, leading to the abnormal function of specific mitochondrial pathways. Mitochondrial dysfunction is also a secondary event in more common pathophysiological conditions, such as obesity and metabolic syndrome. In both cases, the improvement and management of mitochondrial homeostasis remain challenging. Here, we show that beta-resorcylic acid (β-RA), which is a natural phenolic compound, competed in vivo with 4-hydroxybenzoic acid, which is the natural precursor of coenzyme Q biosynthesis. This led to a decrease in demethoxyubiquinone, which is an intermediate metabolite of CoQ biosynthesis that is abnormally accumulated in *Coq9^R239X^* mice. As a consequence, β-RA rescued the phenotype of *Coq9^R239X^* mice, which is a model of primary mitochondrial encephalopathy. Moreover, we observed that long-term treatment with β-RA also reduced the size and content of the white adipose tissue (WAT) that is normally accumulated during aging in wild-type mice, leading to the prevention of hepatic steatosis and an increase in survival at the elderly stage of life. The reduction in WAT content was due to a decrease in adipogenesis, an adaptation of the mitochondrial proteome in the kidneys, and stimulation of glycolysis and acetyl-CoA metabolism. Therefore, our results demonstrate that β-RA acted through different cellular mechanisms, with effects on mitochondrial metabolism; as such, it may be used for the treatment of primary coenzyme Q deficiency, overweight, and hepatic steatosis.

## 1. Introduction

Mitochondria are the primary sites of cellular energy production and also have a broad range of metabolic functions. Thus, mitochondrial dysfunction can produce far-ranging, varied, and severe consequences. Mitochondrial dysfunction can be directly caused by mutations in mitochondrial DNA or mutations in nuclear genes that encode mitochondrial proteins, leading to primary mitochondrial diseases. Aside from direct causes, mitochondrial dysfunction can also occur as a secondary event in more common diseases, such as neurodegenerative diseases, obesity, or metabolic syndrome.

One particular case of mitochondrial disease is coenzyme Q10 (CoQ_10_) deficiency syndrome, which can be primarily caused by mutations in genes that encode proteins that are involved in the CoQ_10_ biosynthetic pathway (primary CoQ_10_ deficiency). Primary CoQ_10_ deficiency presents heterogeneous clinical phenotypes depending on the specific mutation in the CoQ biosynthesis pathway [1,2]. Furthermore, especially given the variety of functions of CoQ, multiple pathomechanisms are induced by low levels of CoQ, including declined bioenergetics [1,3,4,5,6], increased oxidative stress [3,4,7,8], disrupted sulfide metabolism [9,10], and defective de novo pyrimidine biosynthesis [11].

CoQ_10_ deficiency can also be induced as a secondary effect of certain drugs [12] and triggered indirectly by other diseases, including multifactorial diseases and disorders that are caused by mutations in genes that are not related to the CoQ_10_ biosynthesis pathways [13,14,15,16]. Metabolic syndrome is a multifactorial disease with secondary mitochondrial dysfunction. The white adipose tissue (WAT) and skeletal muscle from patients and mice with insulin resistance, which is a characteristic that is usually associated with metabolic syndrome, show decreased levels of the CoQ biosynthetic proteins COQ7 and COQ9, leading to reduced CoQ levels in the mitochondria [17].

In experimental cases of CoQ_10_ deficiency, the levels of CoQ_10_ in blood, cells, and tissues could be increased by exogenous CoQ_10_ supplementation. However, CoQ_10_ has very low absorption and bioavailability when it is orally administrated, and a very low proportion of this exogenous CoQ_10_ can reach the mitochondria of the cells in most tissues [18,19]. Thus, hydroxybenzoic acid derivatives (HBAs) were proposed as an alternative strategy to attenuate CoQ_10_ deficiency since they were shown to modulate the endogenous CoQ biosynthetic pathway [20]. HBAs constitute a group of natural phenolic compounds that are present in plants with a general structure of the C6–C1 type that is derived from benzoic acid. Variable positioning of hydroxyl and methoxy groups on the aromatic ring produce several different compounds, such as 2-hydroxybenzoic acid (or salicylic acid), 4-hydroxybenzoic acid (4-HB), 2,4-dihydroxybenzoic acid (2,4-diHB, or β-resorcylic acid (β-RA)), and 4-hydroxy-3-methoxybenzoic acid (or vanillic acid (VA)). Interestingly, β-RA has a hydroxyl group that is incorporated into the benzoic ring during CoQ biosynthesis. This hydroxylation step is catalyzed by COQ7, which uses demethoxyubiquinone (DMQ) as a substrate and requires the COQ9 protein for its normal function and stability [6]. As a consequence, the administration of high doses of β-RA bypasses the defects in the COQ7 reaction, leading to a dramatic increase in the survival of *Coq7* conditional knockout mice and the *Coq9^R239X^* mice due to increased levels of CoQ and/or to decreased levels of DMQ in the kidneys, heart, skeletal muscle and intestine [21,22,23]. In *Coq9^R239X^* mice, which is a model of mitochondrial encephalopathy due to the accumulation of DMQ and the reduced levels of CoQ, these biochemical changes resulted in significant improvements in encephalopathic features, such as astrogliosis and spongiosis [22]. Similarly, supplementation with high doses of β-RA to podocyte-specific *Coq6* or *Adck4* (*Coq8b*) knockout mice prevented renal dysfunction and increased survival, although the effect of β-RA on CoQ metabolism in these mouse models was not reported and, therefore, the therapeutic mechanisms of these cases are unknown [24,25]. Additionally, Wang and colleagues reported that β-RA decreased the body weight of wild-type mice and increased survival in animals at the middle-age and elderly stages of life, but the mechanisms behind these observations remain to be elucidated. Consequently, these results in the *Coq6* and *Adck4* mouse models and in wild-type mice suggest that β-RA may work through additional unidentified mechanisms.

Here, we evaluated whether a lower dose of β-RA, which may increase its translational potentiality, leads to therapeutic outcomes in the encephalopathic *Coq9^R239X^* mice and whether that effect is mainly due to β-RA interference in CoQ metabolism. Additionally, we tested whether β-RA could be a useful agent to treat the fat accumulation that is linked to aging.

## 2. Materials and Methods

### 2.1. Animals and Treatments

*Coq9*^+/+^ and *Coq9^R239X^* mice were used in the study, both of which harbored a mix of C57BL/6N and C57BL/6J genetic backgrounds. The *Coq9^R239X^* mouse model (MGI: 5473628) was previously generated and characterized [1,6,10]. All animal manipulations were performed according to a protocol that was approved by the Institutional Animal Care and Use Committee of the University of Granada (procedures numbers 18/02/2019/016 18 February 2019 and 16/09/2019/153 16 September 2019) and were in accordance with the European Convention for the Protection of Vertebrate Animals Used for Experimental and Other Scientific Purposes (CETS #123) and the Spanish law (R.D. 53/2013). Mice were housed in the Animal Facility of the University of Granada under an SPF zone with lights on at 7:00 AM and off at 7:00 PM. Mice had unlimited access to water and rodent chow (SAFE^®^ 150, which provided 21, 12.6 and 66.4% of energy from proteins, lipids, and nitrogen-free extracts, respectively). Unless stated otherwise, the analytical experiments were completed on animals at 3 or 18 months of age.

β-Resorcylic acid (β-RA) (Merck Life Science S.L.U, Madrid, Spain) was given to the mice in the chow at a concentration of 0.33% (*w/w*). For some experiments, a concentration of 1% (*w/w*) β-RA was used for two months [22]. A mix of β-RA and 4-HB (at a concentration of 0.5% of each one) was also provided in the chow for particular experiments. Mice began receiving the assigned treatments at 1 month of age, and the analyses were performed at the age indicated for each case. Animals were randomly assigned to experimental groups. Data were randomly collected and processed. 

The body weights were recorded once a month. To weigh the skeletal muscle, mice were sacrificed at 18 months of age and the *gastrocnemius* and *vastus lateralis* were dissected and weighed on a laboratory scale. To weigh the WAT, mice were sacrificed at 18 months of age, and the epididymal, mesenteric, and inguinal WATs were dissected and weighed on a laboratory scale. 

The motor coordination was assessed at different months of age using the rotarod test by recording the length of time that mice could remain on the rod (“latency to fall”), rotating at a rate of 4 rpm, accelerating to 40 rpm in 300 s. Muscle strength was assessed using a computerized grip strength meter (Model 47200, Ugo-Basile, Varese, Italy). The experimenter held the mouse gently by the base of the tail, allowing the animal to grab the metal bar with the forelimbs before being gently pulled until it released its grip. The peak force of each measurement was automatically recorded by the device and expressed in grams (g). The hindlimb grip strength of each mouse was measured in duplicate with at least 1 min between measurements [1].

### 2.2. Cell Culture and Cell Assays

3T3-L1 preadipocytes (ECACC #: 86052701; lot CB 2618) were obtained from the cell bank of the University of Granada and maintained in DMEM containing 10% fetal calf serum (FCS) in a humidified atmosphere of 5% CO_2_ at 37 °C. The differentiation of the preadipocytes was induced 2 days post-confluence (day 0) following the manufacturer’s instructions (DIF001-1KT; Merck Life Science S.L.U, Madrid, Spain)) via the addition of 0.5 mM 3-isobutyl-1-methylxanthine (IBMX), 1 μM dexamethasone, and 10 μg/mL insulin (multiple daily insulin (MDI)) for 2 days. Subsequently, the culture medium was changed to DMEM and 10% fetal bovine serum (FBS) containing insulin. After 2 days, the medium was replaced with DMEM and 10% FBS, and the cells were incubated for a further 2 days until the cells were harvested to be used in the experiments described below. 

C2C12 myocytes (ECACC #: 91031101; lot 08F021) were obtained from the cell bank at the University of Granada and maintained in DMEM containing 10% FBS in a humidified atmosphere of 5% CO_2_ at 37 °C. The differentiation of the preadipocytes was induced 1-day post-confluence (day 0) by changing to a 1% FBS medium. Subsequently, the culture medium was changed to DMEM and 1% FBS. The medium was changed every other day and the cells were harvested to be used in the experiments described below.

In both cell types, namely, 3T3-L1 and C2C12, each assay was carried out in one of three experimental conditions: proliferative, differentiative, or proliferative + differentiative. Proliferative conditions were developed in both types of cells after cell splitting, and cells were collected upon reaching the confluency at day 7. Differentiative conditions were initiated in both cell types when the cells reached confluency. In 3T3-L1 cells, the differentiation was induced with the differentiation medium described above. In the C2C12 cells, differentiation was induced in a medium with 1% FBS, as described above. The cells were collected on day 7. Proliferative + differentiative conditions combined both procedures in the same experiment. β-RA was added at a final concentration of 1 mM every other day in each experimental condition.

To visualize the lipid droplets, the 3T3-L1 cells were fixed in formalin and stained with Oil Red solution on days 2, 4, and 6 in both the proliferative and proliferative + differentiative conditions.

Cell viability and proliferation were quantified on day 7 using a Vybrant MTT Cell Proliferation Assay Kit according to the manufacturer’s instructions (Thermofisher, Madrid, Spain). Absorbance was measured at 450 nm on a microplate reader (Powerwave ×340 spectrophotometer; Biotek, Winooski, VT, USA).

### 2.3. Histology and Immunohistochemistry

Tissues were fixed in formalin and embedded in paraffin. Multiple sections (4 μm thickness) were deparaffinized with xylene and stained with hematoxylin and eosin (H&E) (Merck Life Science S.L.U, Madrid, Spain), Masson’s trichrome, or Oil Red (Merck Life Science S.L.U, Madrid, Spain). Immunohistochemistry was carried out on the same sections using the following primary antibodies: glial fibrillary acidic protein or anti-GFAP (glial fibrillary acidic protein) (MAB360; Millipore, Madrid, Spain). The Dako Animal Research Kit for mouse primary antibodies (Dako, Agilent Technologies, Madrid, Spain) was used for the qualitative identification of antigens by light microscopy. Sections were examined at 40–400 magnifications with a Nikon Eclipse Ni-U microscope (Werfen, Madrid, Spain), and the images were scanned under equal light conditions with the NIS-Elements Br computer software (Werfen, Madrid, Spain).

### 2.4. Plasma and Urine Analysis

Blood samples were collected in K_3_-ethylenediaminetetraacetic acid (EDTA) tubes (Kima, VWR, Barcelona, Spain) using a goldenrod lancet and the submandibular vein of each mouse as a puncture site. The plasma was extracted from blood samples via centrifugation at 4500× *g* for 10 min at 4 °C. Biochemical analyses of the urine and plasma were developed in a biochemical analyzer Bs-200 (Shenzhen Mindray Bio-Medical Electronics Co., Ltd., Shenzhen, China) using reagents from Spinreact.

The NEFAS concentration was quantified using the Free Fatty Acid Quantitation Kit (MAK044) according to the technical bulletin (Merck Life Science S.L.U, Madrid, Spain). The results were expressed in nanograms per microliter.

The insulin concentration was quantified using the Mouse INS ELISA Kit (EM0260) according to the manufacturer’s instructions (FineTest, Labclinics, Barcelona, Spain). The results were expressed in picograms per milliliter.

The Glucagon concentration was quantified using the Mouse GC ELISA Kit (EM0562) according to the manufacturer’s instructions (FineTest, Labclinics, Barcelona, Spain).The results were expressed in picograms per milliliter.

### 2.5. Mitochondrial Proteomics Analysis

Both the *Coq9*^+/+^ mice and *Coq9*^+/+^ mice that were given the 1% β-RA supplementation were sacrificed, and the kidneys were removed and washed in a saline buffer. The tissues were chopped with scissors in 3 mL HEENK (10 mM 4-(2-hydroxyethyl)-1-piperazineethanesulfonic acid (HEPES), 1 mM EDTA, 1 mM ethylene glycol-bis(β-aminoethyl ether)-N,N,N′,N′-tetraacetic acid (EGTA), 10 mM NaCl, 150 mM KCl, pH 7.1, 300 mOsm/L) (Merck Life Science S.L.U, Madrid, Spain) containing 1 mM phenylmethanesulfonyl fluoride (PMFS) (Merck Life Science S.L.U, Madrid, Spain) (from 0.1 M stock in isopropanol) and 1× protease inhibitor cocktail (Pierce). The tissues were homogenized with a 3 mL Dounce homogenizer (5 passes of a tight-fitting Teflon piston). Each obtained homogenate was rapidly subjected to standard differential centrifugation methods until a mitochondrial pellet was obtained, as previously described [26]. Briefly, the homogenate was centrifuged at 600× *g* for 5 min at 4 °C (twice), and the resultant supernatant was centrifuged at 9000× *g* for 5 min at 4 °C. The final pellet, corresponding to a crude mitochondrial fraction, was resuspended in 500 μL of HEENK medium without PMSF or protease inhibitor [26]. The protein concentration was determined (using Bradford dye, Bio-Rad, Madrid, Spain) and a Shimadzu spectrophotometer, resulting in approximately 3 mg protein for renal mitochondria and 1.5 mg for cerebral mitochondria. To verify the content of the mitochondrial fraction, complex IV activity was determined using optical absorption of the difference spectrum at 550 nm, as previously described [10].

The purified mitochondria were spun down to remove the previous buffer, and lysis buffer (1% sodium deoxycholate SDC in 100 mM Tris at pH 8.5) was added to the pellets. The samples were boiled for 5 min at 99 °C to denature all the proteins and then sonicated using microtip probe sonication (Hielscher UP100H Lab Homogenizer, Hielscher Ultrasonics GmbH, Teltow, Germany) for 2 min with pulses of 1 s on and 1 s off at 80% amplitude. The protein concentration was estimated using a bicinchoninic acid assay (BCA) and 200 µg were taken from each sample. Then, 10 mM tris(2-carboxyethyl)phosphine and 40 mM chloroacetamide (final concentration) at 56 °C were added to each of these 200 µg samples for 10 min to reduce and alkylate the disulfide bridges. After this step, samples were digested with LysC (FUJIFILM Wako Chemicals Europe GmbH, Neuss, Germany) in an enzyme/protein ratio of 1:100 (*w/w*) for 1 h, followed by a trypsin digest (Promega, Leiden, The Netherlands) 1:50 (*w/w*) overnight. Protease activity was quenched with trifluoroacetic acid (TFA) to a final pH of ~2. Samples were then centrifuged at 5000× *g* for 10 min to eliminate the insoluble SDC, and loaded on an OASIS HLB (Waters Chromatography Europe, Etten-Leur, The Netherlands) 96-well plate. Samples were washed with 0.1% TFA, eluted with a 50/50 acetonitrile (ACN) and 0.1% TFA, dried using a SpeedVac (Eppendorf, Hamburg, Germany), and resuspended in 2% formic acid prior to the MS analysis. From each sample, 5 µg were taken and pooled in order to be used for quality control for MS (1 QC was analyzed every 12 samples) and to be fractionated at a high pH for the match between runs.

All samples with the QC and 7 high-pH fractions were acquired using a UHPLC 1290 system (Agilent Technologies, Santa Clara, CA, USA) that was coupled online to a Q Exactive HF mass spectrometer (Thermo Scientific, Bremen, Germany). Peptides were first trapped (Dr. Maisch Reprosil C18, 3 μm, 2 cm × 100 μm) prior to separation on an analytical column (Agilent Poroshell EC-C18, 2.7 μm, 50 cm × 75 μm). Trapping was performed for 5 min in solvent A (0.1% *v/v* formic acid in water), and the gradient was as follows: 10–40% solvent B (0.1% *v/v* formic acid in 80% *v/v* ACN) over 95 min, 40–100% B over 2 min, then the column was cleaned for 4 min and equilibrated for 10 min (flow was passively split to approximately 300 nL/min). The mass spectrometer was operated in a data-dependent mode. Full-scan MS spectra in the range of m/z 300–1600 Th were acquired in the Orbitrap at a resolution of 120,000 after accumulation to a target value of 3 × 10^6^ with a maximum injection time of 120 ms. The 15 most abundant ions were fragmented with a dynamic exclusion of 24 s. HCD fragmentation spectra (MS/MS) were acquired in the Orbitrap at a resolution of 30,000 after accumulation to a target value of 1 × 10^5^ with an isolation window of 1.4 Th and a maximum injection time of 54 ms.

All raw files were analyzed with MaxQuant v1.6.10 software (Martinsried, Germany) [27] using the integrated Andromeda Search engine and searched against the mouse UniProt Reference Proteome (November 2019 release with 55,412 protein sequences) with common contaminants. Trypsin was specified as the enzyme, allowing up to two missed cleavages. Carbamidomethylation of cysteine was specified as fixed modification and protein N-terminal acetylation, oxidation of methionine, and deamidation of asparagine were considered variable modifications. We used all the automatic settings and activated the “match between runs” (time window of 0.7 min and alignment time window of 20 min) and LFQ with standard parameters. The files generated by MaxQuant were opened in Perseus for the preliminary data analysis: the LFQ data were first transformed in log2, then the identifications that were present in at least N (3/5) biological replicates were kept for further analysis; missing values were then imputed using the standard settings of Perseus. Ingenuity pathway analysis (IPA) was used to identify the changes in metabolic canonical pathways and their z-score predictions [28].

### 2.6. Sample Preparation and Western Blot Analysis in Tissues and Cells

For the Western blot analyses, a glass Teflon homogenizer was used to homogenize the mouse kidney, liver, skeletal muscle, and WAT samples at 1100 rpm in a T-PER^®^ buffer (Thermo Scientific, Madrid, Spain) with a protease and phosphatase inhibitor cocktail (Pierce, Fisher Scientific, Madrid, Spain). Homogenates were sonicated and centrifuged at 1000× *g* for 5 min at 4 °C, and the resultant supernatants were used for the Western blot analysis. For the Western blot analyses of the cells, the pellets containing the cells were re-suspended in RIPA buffer with a protease inhibitor cocktail. About 30 μg of protein from the sample extracts were electrophoresed in 12% Mini-PROTEAN TGXTM precast gels (Bio-Rad) using the electrophoresis system mini-PROTEAN Tetra Cell (Bio-Rad). Proteins were transferred onto PVDF 0.45 μm membranes using a Trans-Blot Cell (Bio-Rad) and probed with target antibodies. Protein–antibody interactions were detected using peroxidase-conjugated horse anti-mouse, anti-rabbit, or anti-goat IgG antibodies and Amersham ECLTM Prime Western Blotting Detection Reagent (GE Healthcare, Buckinghamshire, UK). Band quantification was carried out using an Image Station 2000R (Kodak, Madrid, Spain) and Kodak 1D 3.6 software (Kodak, Madrid, Spain). Protein band intensity was normalized to VDAC1 for mitochondrial proteins and to GAPDH or β-actin for cytosolic proteins. The data were expressed in terms of the percent relative to wild-type mice or control cells.

The following primary antibodies were used: anti-ALDH1B1 (15560-1-AP, Proteintech, Manchester, UK), anti-GSK3B (22104-1-AP, Proteintech, Manchester, UK), anti-EHHADH (sc-393123, Santa Cruz, Heidelberg, Germany), anti-ACADM (ab110296, Abcam, Cambridge, UK), anti-SKP2 (15010-AP, Proteintech, Manchester, UK), anti-P27 (25614-1-AP, Proteintech, Manchester, UK), anti-Cyc A2 (18202-1-AP, Proteintech, Manchester, UK), anti-β-ACTIN (sc-47778, Santa Cruz, Heidelberg, Germany), anti-PPARγ (MA5-14889, Thermo Scientific, Madrid, Spain), anti-PPARδ (PA1-823A, Thermo Scientific, Madrid, Spain), anti-AMPK (#2532, Cell Signaling, Danvers, MA, USA), anti-P-AMPK (#2531, Cell Signaling, Danvers, MA, USA), anti-ULK1 (#8054, Cell Signaling, Danvers, MA, USA), anti-P-ULK1 (#5869, Cell Signaling, Danvers, MA, USA), anti-ACC (#3676, Cell Signaling, Danvers, MA, USA), and anti-P-ACC (#11818, Cell Signaling, Danvers, MA, USA).

### 2.7. Quantification of CoQ_9_ and CoQ_10_ Levels in Mice Tissues and 3T3-L1 Cells

After the lipid extraction from homogenized tissues and cultured cells, CoQ_9_ and CoQ_10_ levels were determined via reversed-phase HPLC coupled to electrochemical detection, as previously described [1,6]. The results were expressed in nanograms of CoQ per milligram of protein.

### 2.8. CoQ-Dependent Respiratory Chain Activities

Coenzyme Q-dependent respiratory chain activities were measured in tissue samples of brain, kidney, skeletal muscle, and heart. Tissue samples were homogenized in a CPT medium (0.05 M Tris-HCl, 0.15 M KCl, pH 7.5) at 1100 rpm in a glass Teflon homogenizer. Homogenates were sonicated and centrifuged at 600× *g* for 20 min at 4 °C, and the obtained supernatants were used to measure the CoQ-dependent respiratory chain activities (CI  +  III and CII  +  III), as previously described [22]. 

### 2.9. Metabolic Assays in Tissues

Phosphofructokinase enzyme activity was measured using a kit from Merck Life Science S.L.U. (Madrid, Spain) (Phosphofructokinase Activity Colorimetric Assay Kit MAK093) according to the manufacturer’s instructions. The enzyme activity was expressed in micromoles per minute per milligram of protein.

Pyruvate kinase enzyme activity was measured using a kit from Merck Life Science S.L.U. (Madrid, Spain) (Pyruvate kinase Activity Colorimetric Assay Kit MAK072) according to the manufacturer’s instructions. The enzyme activity was expressed in picomoles per minute per milligram of protein.

The G3P concentration was quantified using the Glycerol-3-Phosphate Assay Kit (MAK207) according to the technical bulletin (Merck Life Science S.L.U, Madrid, Spain). The concentration of G3P was expressed in nanograms per microgram of protein.

The BHB concentration was quantified using the Beta-Hydroxybutyrate Assay Kit (MAK041) according to the technical bulletin (Merck Life Science S.L.U, Madrid, Spain). The concentration of BHB was expressed in nanograms per microgram of protein.

### 2.10. Mitochondrial Respiration

Mitochondrial isolation from the brain and the kidneys was performed as previously described [22,29]. To isolate fresh mitochondria, mice were sacrificed and the organs were extracted rapidly and put on ice. Brain was homogenized (1:10, w/v) in a respiration buffer C (0.32 M sucrose, 1 mM EDTA-K+, 10 mM Tris-HCl, pH 7.4) at 500 rpm at 4 °C in a glass Teflon homogenizer. The homogenate was centrifuged at 13,000× *g* for 3 min at 4 °C. The supernatant (s1) was kept on ice and the pellet was re-suspended in 5 mL of buffer A and centrifuged at 13,000× *g* for 3 min at 4 °C. The subsequent supernatant (s2) was combined with s1 and centrifuged at 21,200× *g* for 10 min at 4 °C. The mitochondrial pellet of this step was re-suspended in a 0.85 mL extraction buffer A containing 15% Percoll, poured into ultracentrifuge tubes containing a Percoll gradient formed by 1 mL 40% Percoll and 1 mL 23% Percoll in buffer A, and centrifuged at 63,000× *g* for 30 min at 4 °C. Pure mitochondria, corresponding to a fraction between 23 and 40% Percoll, were collected and washed twice with 1 mL of buffer A at 10,300× *g* for 10 min at 4 °C. The mitochondrial pellets were suspended in MAS 1× medium. Kidney was homogenized (1:10, w/v) in a respiration buffer A (250 mM sucrose, 0.5 mM Na_2_EDTA, 10 mM Tris, and 1 % free fatty acid albumin) at 800 rpm in a glass Teflon homogenizer. Then, the homogenate was centrifuged at 500× *g* for 7 min at 4 °C and the supernatant was centrifuged at 7800× *g* for 10 min at 4 °C. The pellet was then resuspended in respiration buffer B (250 mM sucrose, 0.5 mM Na2EDTA, and 10 mM Tris) and an aliquot was used for the protein determination. The remaining sample was then centrifuged at 6000× *g* for 10 min at 4 °C. The pellet was resuspended in buffer A and centrifuged again at 6000× *g* for 10 min at 4 °C. The final crude mitochondrial pellet was re-suspended in MAS 1× medium [70 mM sucrose, 220 mM mannitol, 10 mM KH_2_PO_4_, 5 mM MgCl_2_, 2 mM HEPES, 1 mM EGTA and 0.2% (w/v) fatty acid-free BSA, pH 7.2, all from Merck Life Science S.L.U, Madrid, Spain].

Mitochondrial respiration was measured with an XF^e24^ Extracellular Flux Analyzer (Seahorse Bioscience, Agilent Technologies, Madrid, Spain) [22,29,30]. The mitochondria were first diluted in cold MAS 1× for plating (3.5 μg/well in brain; 2 μg/well in kidney). Next, 50 μL of mitochondrial suspension was delivered to each well (except for background correction wells) while the plate was on ice. The plate was then centrifuged at 2000× *g* for 10 min at 4 °C. After centrifugation, 450 μL of MAS 1× + substrate (10 mM succinate, 2 mM malate, 2 mM glutamate, and 10 mM pyruvate) was added to each well. Respiration by the mitochondria was sequentially measured in a coupled state with the substrate present (basal respiration or state 2) followed by state 3o (phosphorylating respiration, in the presence of ADP and substrates). State 4 (non-phosphorylating or resting respiration) was measured after the addition of oligomycin when all the ADP was consumed, and then the maximal uncoupler-stimulated respiration was measured using carbonyl cyanide-p-trifluoromethoxyphenylhydrazone (FCCP) (state 3u). Injections were as follows: port A, 50 μL of 40 mM ADP (4 mM final); port B, 55 μL of 30 μg/mL oligomycin (3 μg/mL final); port C, 60 μL of 40 μM FCCP (4 μM final); and port D, 65 μL of 40 μM antimycin A (4 μM final). All data were expressed in picomoles per minute per milligram of protein. The respiratory control ratio (RCR) was calculated using the highest OCR point in state 3o and the lowest point in state 4 Merck Life Science S.L.U. (Madrid, Spain) was the manufacturer of the succinate, malate, glutamate, pyruvate, ADP, oligomycin, FCCP, and antimycin

### 2.11. Quantification of β-RA and 4-HB Levels in Mice Tissues

Tissues from the mice were homogenized in water. The homogenate samples were then treated with a solution of methanol/water (80:20, *v/v*), shook for 1 min, sonicated for 15 min, and then centrifuged at 5000× *g* for 25 min at 4 °C. 

The supernatants were analyzed using a Thermo Scientific™ UltiMate™ 3000 UHPLC system (Waltham, MA, USA), consisting of an UltiMate™ 3000 UHPLC RS binary pump and an UltiMate™ 3000 UHPLC sample manager coupled to a Thermo Scientific™ Q Exactive™ Focus Hybrid Quadrupole-Orbitrap™ detector of a mass spectrometer (MS/MS) with electrospray ionization in negative mode (Waltham, Massachusetts, United States). The analytical separation column was a Hypersil GOLD™ C18, 3 μm, 4.6 mm × 150 mm column (Thermo Scientific, Madrid, Spain) and the flow rate was 0.6 mL/min. The mobile phase consisted of two solutions: eluent A (H_2_O + 0.1% formic acid, MS grade, Thermo Scientific, Madrid, Spain) and eluent B (acetonitrile + 0.1% formic acid, MS grade, Thermo Scientific, Madrid, Spain). Samples were eluted over 30 min with the following gradient: 0 min, 95% eluent A; 0–25 min, 70% eluent A; 25–25.1 min, 95% eluent A; 25.1–30 min, 95% eluent A. The capillary and auxiliary gas temperatures were set at 275 and 450 °C, respectively. The sheath gas flow rate used was at 55 arbitrary units, the auxiliary gas flow rate used was at 15 arbitrary units, and the sweep gas flow was used at 3 arbitrary units. Mass spectrometry analyses were carried out in full scan mode between 110 and 190 uma. To quantify the levels of 4-HB (Merck Life Science S.L.U, Madrid, Spain) and β-RA, we used a standard curve with both compounds at concentrations of 100, 10, and 1 ng/mL. 

### 2.12. Statistical Analysis

The number of animals in each group was calculated in order to detect gross ~60% changes in the biomarker measurements (based upon alpha = 0.05 and power of beta = 0.8). We used the application available at http://www.biomath.info/power/index.htm accessed on 14 September 2021. Animals were genotyped and randomly assigned to experimental groups in separate cages by the technician of the animal facility. Most statistical analyses were performed using the Prism 9 scientific software. Data are expressed as the mean ± SD of five to ten experiments per group. A one-way ANOVA with Tukey’s post hoc test was used to compare the differences between the three experimental groups. Studies with two experimental groups were evaluated using the Mann–Whitney (nonparametric) test. A *p*-value of <0.05 was considered to be statistically significant. The survival curve was analyzed using log-rank (Mantel–Cox) and the Gehan–Breslow–Wilcoxon tests. The statistical tests that were used for the transcriptomics and proteomics analyses are described in their respective sections.

## 3. Results

### 3.1. β-RA Induced Phenotypic and Morphological Benefits against Both Age-Related Obesity and Mitochondrial Encephalopathy due to CoQ Deficiency

β-RA was incorporated into the chow of both wild-type and *Coq9^R239X^* mice at a concentration of 0.33% (*w/w*), which gave a dose of 0.4–0.7 g/kg b.w./day, considering the animal food intake, which was similar in all groups (Figure 1A–C). This low dose of β-RA improved the survival of *Coq9^+/+^* mice at the old stage of life (Figure 1D,E), where 87% of the treated *Coq9^+/+^* mice survived compared with 62% of the untreated mice. However, the survival curve became similar to the survival curve of untreated animals after 28 months of age. Similarly, the low-dose treatment of β-RA also improved the survival of *Coq9^R239X^* mice (Figure 1D), and we even observed a maximal lifespan higher than the maximal lifespan reported when *Coq9^R239X^* mice were treated with a high dose of β-RA [22].

The encephalopathic features of *Coq9^R239X^* mice result in characteristics of lower locomotor activity and increased uncoordination. However, the *Coq9^R239X^* mice improved after β-RA administration compared to the untreated *Coq9^R239X^* mice. The treatment did not significantly affect the results of the rotarod test in wild-type animals (Figure 1F,G). Both the *Coq9^+/+^* and *Coq9^R239X^* mice treated with β-RA had a healthy appearance (Movies S1 and S2).

The body weights were significantly reduced in both male and female *Coq9^+/+^* mice after one month of treatment, reaching a maximal weight of about 28 g in males and 23 g in females at seven months of age. These weights were then maintained throughout the remaining life of the animals (Figure 1H,I) (Appendix A). Curiously, the treatment with β-RA slightly increased the body weights of the *Coq9^R239X^* mice, which usually weighed less than their untreated *Coq9^+/+^* littermates (Figure 1H,I). Consequently, both treated *Coq9^+/+^* and treated *Coq9^R239X^* mice had similar body weights. The reduced body weight in *Coq9^+/+^* mice after the β-RA treatment was mainly caused by the prevention of accumulation of WAT (Figure 1J,K,N,O) while still preserving the content, weight, and strength of the skeletal muscle (Figure 1L–O and Appendix A).

The most notable histopathological features of CoQ_10_ deficiency in the *Coq9^R239X^* mice were cerebral spongiosis and reactive astrogliosis (Figure 2(A1–D1)), together with the reduced body weight due to, at least in part, to the decreased content in WAT (Appendix A(A1,B1)). Low-dose β-RA supplementation in the *Coq9^R239X^* mice for two months decreased the characteristic spongiosis (marked by an arrow, Figure 2(E1,F1)) and reactive astrogliosis, determined using the GFAP-positive cells (marked by an arrow, Figure 2(G1,H1)), with no changes in the liver (Appendix A(C1–J1)). These results were similar to the therapeutic effect that was previously reported with a higher dose [22]. In *Coq9^+/+^* mice, β-RA supplementation for two months did not produce significant morphological alterations in the brain (Figure 2(I1–P1)), liver (Appendix A(K1–M1) and (Q1–S1)), kidneys (Appendix A(N1–P1) and (T1–V1), spleen (Appendix A), heart (Appendix A), or small intestine (Appendix A), and the blood and urine markers of the renal and hepatic functions did not reveal any abnormality (Appendix A).

At 18 months of age, the livers of both male and female wild-type mice showed features of steatosis (Figure 2(Q1–X1) and Figure 2(G2,H2)). Chronic supplementation with β-RA dramatically reduced the signs of hepatic steatosis (Figure 2(Y1–F2) and Figure 2(G2,H2)). Non-alcoholic hepatic steatosis is frequently associated with fat accumulation. Consequently, the epididymal WAT showed characteristics of hypertrophy in both the male and female *Coq9^+/+^* mice at 18 months of age (Figure 2(I2–L2) and Figure 2(Q2–T2)), with adipocytes that were bigger in size and lower in number per area. β-RA supplementation suppressed the epididymal WAT hypertrophy in both the male and female *Coq9^+/+^* mice at 18 months of age (Figure 2(M2–P2) and Figure 2(Q2–T2)). At 18 months of age, no major alterations were found in the brains or kidneys (Appendix A).

### 3.2. β-RA Led to Bioenergetics Improvement in Coq9^R239X^ Mice through Its Direct Participation in the CoQ Biosynthetic Pathway

The decrease in DMQ_9_ was previously reported as the main therapeutic mechanism of a high dose of β-RA in the treatment in *Coq9^R239X^* mice, although the effects in the CoQ biosynthetic pathway in wild-type animals were not evaluated [22]. Thus, we evaluated whether a lower dose of β-RA interferes with CoQ biosynthesis in both *Coq9^+/+^* and *Coq9^R239X^* mice. In *Coq9^+/+^* mice, β-RA induced very mild changes in the tissue levels of CoQ_9_, CoQ_10_, and DMQ_9_ (Figure 3(A1–L1), Appendix A). The levels of CoQ_9_ were similar in the brain, kidneys, liver heart, and WAT of untreated and treated wild-type mice, whilst in skeletal muscle, the β-RA induced a mild reduction in the levels of CoQ_9_ (Figure 3(A1–D1), Appendix A). DMQ_9_ was undetectable in the tissues of untreated wild-type mice, and β-RA supplementation induced the accumulation of very low levels of DMQ_9_ in the kidneys, liver, skeletal muscle, and WAT, but not in the brain or heart (Figure 3(I1–L1), Appendix A). Consequently, the ratio DMQ_9_/CoQ_9_ was not significantly altered in *Coq9^+/+^* mice treated with β-RA, as it was observed in the untreated *Coq9^R239X^* mice (Figure 3(M1–P1)). In *Coq9^R239X^* mice, β-RA administration induced a mild increase in CoQ_9_ in the kidneys (Figure 3(B1) and Appendix A) compared with untreated *Coq9^R239X^* mice. However, the levels of CoQ_9_ did not change in the brain, liver, skeletal muscle, or heart of *Coq9^R239X^* mice after the β-RA treatment (Figure 3A1, Figure 3C1, Figure 3D1 and Appendix A). Remarkably, the levels of DMQ_9_ and, consequently, the DMQ_9_/CoQ_9_ ratio, were significantly decreased in the kidneys (Figure 3(J1,N1) and Appendix A), liver (Figure 3(K1,O1)), skeletal muscle (Figure 3L1,P1), and heart (Appendix A) of the *Coq9^R239X^* mice treated with β-RA compared with the untreated *Coq9^R239X^* mice. However, β-RA did not reduce the levels of DMQ_9_ or the DMQ_9_/CoQ_9_ ratio in the brain of the *Coq9^R239X^* mice (Figure 3(I1,M1)), as it was also reported in the treatment with the higher dose of β-RA [22]. Therefore, the effect of β-RA on CoQ metabolism in the *Coq9^R239X^* mice in this study was similar to the effect previously reported with a higher dose of β-RA, i.e., a decrease in the DMQ/CoQ ratio in peripheral tissues [22].

The tissue-specific reduction in the levels of DMQ_9_ in *Coq9^R239^*^X^ mice seemed to correlate with the increase in β-RA since the levels of β-RA were higher in the kidneys (Figure 3R1), liver (Figure 3(S1)), skeletal muscle (Figure 3(T1)), and heart (Appendix A) than in the brain (Figure 3Q1) of *Coq9^R239^*^X^ mice. The levels of 4-HB, the natural precursor for CoQ biosynthesis, did not increase in response to the treatment with β-RA in any tissue of either the *Coq9^+/+^* or *Coq9^R239X^* mice (Figure 3U1, Figure 3V1, Figure 3W1, Figure 3X1 and Appendix A).

Bioenergetically, the treatment with β-RA did not produce any changes in the brain in either the *Coq9^+/+^* or *Coq9^R239X^* mice (Figure 3(Y1,C2) and Appendix A), but it did increase the activities of complexes I + III and II + III (Figure 3(Z1,D2)) and mitochondrial respiration (Appendix A) in the kidneys of the treated *Coq9^R239X^* mice compared to the untreated *Coq9^R239X^* mice. These data are comparable to those reported for the treatment with the high dose of β-RA [22], suggesting that the decrease in the DMQ/CoQ ratio was responsible for the bioenergetics improvement. Other tissues did not experience major changes in mitochondrial bioenergetics in *Coq9^+/+^* or *Coq9^R239X^* mice (Figure 3(Y1–G2) and Appendix A).

Because β-RA is an analog of 4-HB, its effects at reducing DMQ_9_ in *Coq9^R239X^* mice were most likely due to its competition with 4-HB when entering the CoQ biosynthetic pathway through the activity of COQ2. To investigate this hypothesis, we supplemented the *Coq9^+/+^* and *Coq9^R239X^* mice with an equal amount of 4-HB and β-RA incorporated into the chow. Because COQ2 has more of an affinity for 4-HB than for β-RA, in conditions of equal amounts of both compounds, COQ2 will preferably use 4-HB. Accordingly, the co-administration of 4-HB and β-RA suppressed the mild inhibitory effect of β-RA over CoQ_9_ biosynthesis in the skeletal muscle (Figure 4D) and CoQ_10_ biosynthesis in the brain, kidneys, and liver (Figure 4F–H) of the *Coq9^+/+^* mice (compare with Figure 3). Moreover, CoQ_9_ increased in the brain (Figure 4A) and the kidneys (Figure 4B) of the *Coq9^+/+^* mice treated with the combination of 4-HB and β-RA compared to the untreated *Coq9^+/+^* mice. In the *Coq9^R239X^* mice, the untreated and treated groups showed similar levels of both CoQ_9_ (Figure 4A–E) and CoQ_10_ (Figure 4F–J) in all tissues. Importantly, the reduction in the levels of DMQ_9_ and the DMQ_9_/CoQ_9_ ratio induced by β-RA (Figure 3, Appendix A) in the *Coq9^R239X^* mice seemed to be suppressed by the co-administration of 4-HB and β-RA (Figure 4K–T). Consequently, the co-administration of 4-HB and β-RA suppressed the increase in survival of the *Coq9^R239X^* mice that was found after the treatment with β-RA alone (Figure 4U). Together, these data demonstrated that β-RA acted therapeutically in the *Coq9^R239X^* mice by entering the CoQ biosynthetic pathway, leading to a reduction in the levels of DMQ_9_. 

### 3.3. A Metabolic Switch in Wild-Type Animals Contributed to the Effects of β-RA in Reducing WAT

Since the interference of β-RA in CoQ metabolism in wild-type mice was very mild, the profound reduction in WAT was not likely attributed to CoQ metabolism. Thus, we investigated whether β-RA can target other mitochondrial pathways by performing quantitative proteomics on mitochondrial fractions of kidneys from wild-type mice treated with 1% β-RA for only two months and compare the results to those of kidneys from the untreated wild-type mice (Data File S1). We chose a higher dose to ensure that the effects of the β-RA supplementation were evident. Furthermore, the analysis was done in the kidneys because this tissue maintained the highest levels of β-RA after the supplementation. In the kidneys of the wild-type mice treated with β-RA compared to kidneys of the untreated wild-type mice, 442 mitochondrial proteins were differentially expressed (Data File S2), with 300 proteins being overexpressed and 142 proteins being underexpressed. Canonical metabolic analysis showed enrichment (top 10) of the pathways of fatty acid β-oxidation, acetyl-CoA biosynthesis, the tricarboxylic acid (TCA) cycle, and the 2-ketoglutarate dehydrogenase complex, as well as enrichment of the related branched-chain α-keto acid dehydrogenase complex (Figure 5A). Importantly, the prediction z-score revealed an inhibition of fatty acid β-oxidation and activation of acetyl-CoA biosynthesis and the TCA cycle (Figure 5A), which was consistent with the changes found in the levels of key proteins in these pathways (Figure 5B). Western blotting for the proteins ALDH1B1, GSK3β, EHHADH, and ACADM from the mice fed at 1 or 0.33% β-RA in the diet (Figure 5C,D) validated these findings in the kidneys. Taken together, the results of the mitochondrial proteome analysis suggested that β-RA treatment stimulates the production and use of acetyl-CoA in the kidneys while repressing fatty acid β-oxidation in the kidneys (Figure 5E). Thus, we hypothesized that β-RA supplementation induces glycolysis at the expense of fatty acid β-oxidation. For this, lipolysis may induce an increase in glycerol-3-P (G3P), which may stimulate glycolysis to provide the substrate for acetyl-CoA biosynthesis. Accordingly, the activities of the glycolytic enzymes phosphofructokinase (PFK) and pyruvate kinase (PK) were partially increased with the β-RA treatment (Figure 5F,G). Moreover, G3P were increased with the β-RA treatment (Figure 5H), while the levels of β-hydroxybutyrate (BHB) showed a notable but statistically insignificant increase with the β-RA treatment (Figure 5I).

We performed similar analyses in the liver and skeletal muscle, which are two relevant tissues in the regulation of systemic energy metabolism, to check whether this metabolic switch was a common phenomenon. The levels of the proteins ALDH1B1, GSK3β, EHHADH, and ACADM in the liver and skeletal muscle did not change like the changes observed in the kidneys (Figure 6A–F and Appendix A). However, PFK activity increased with the β-RA treatment in both tissues (Figure 6G and Appendix A), suggesting the activation of glycolysis despite a lack of change of PK activity from the treatment (Figure 6H and Appendix A). Moreover, G3P increased in the liver with the treatment of 1% β-RA, although these levels did not change at the low dose nor in the skeletal muscle with both doses (Figure 6I and Appendix A). In the liver, the levels of BHB showed an observable but statistically insignificant increase with the β-RA treatment (Figure 6J). The levels of *Fgf21*, which is a secretory endocrine factor that can affect systemic glucose and lipid metabolism [31], trended upward with the β-RA treatment (Figure 6K). An increase in BHB levels was also observed in the blood plasma with the treatment of 1% β-RA (Figure 6L). However, the levels of non-esterified fatty acids (NEFA), which are products of lipolysis, were similar in the treated and untreated animals (Figure 6M). Furthermore, the levels of glucagon, insulin, and the insulin/glucagon ratio were similar in the treated and untreated animals, which most likely reflected a homeostatic status with the chronic administration of β-RA (Figure 6N–P). These results suggest that metabolism in the kidneys and, to a lesser extent, the liver contributed to the reduced WAT that was induced by β-RA in wild-type animals.

### 3.4. β-RA Directly Inhibited Adipogenesis

While the metabolic switch in the kidneys and, to a lesser extent, the liver may contribute to the utilization of energetic substrates that prevent the accumulation of WAT, we also wondered whether β-RA directly affects adipocytes. This is important because mitochondrial metabolism was related to the inhibition of preadipocytes proliferation [32,33]. Thus, we treated 3T3-L1 preadipocytes with β-RA. In proliferative conditions, β-RA decreased cell proliferation (Figure 7A,D), most likely due to an increase in p27 (Figure 7B), which is a protein that inhibits the cell cycle progression at G1 [34,35]. We also observed a decrease in CYCA2, which is a protein that promotes the division of the cells [35] (Figure 7B). These changes in p27 and CYCA2 were not observed in differentiated 3T3-L1 cells (Figure 7C) nor in C2C12 myoblasts under both proliferative and differentiative conditions (Appendix A), indicating a specific cell-type effect. Consistently, the 3T3-L1 cells treated with β-RA produced less fat (Figure 7E,F), which was a phenomenon that may have been mediated by the decrease in PPARγ levels (Figure 7G) and the upward trend of PPARδ levels (Figure 7H), which are two receptors that regulate adipogenesis [36,37]. The decreased levels of CoQ_9_ (Figure 7I,J) due to the competitive inhibition of CoQ biosynthesis induced by β-RA in the control cells, which is a fact that was previously reported [1,38], could also contribute to the decreased proliferation and fat production of 3T3-L1 cells [32,39].

Because other HBAs, e.g., salicylic acid or vanillic acid, can activate AMPK [40,41], which is an enzyme that plays a key role in cellular energy homeostasis [42,43], we investigated whether the observed effects of β-RA in WAT were due to the activation of AMPK through its phosphorylation. Thus, we quantified the levels of AMPK and *p*-AMPK, as well as two of its target proteins, ULK1/p-ULK1 and ACC/p-ACC, in the WAT of wild-type mice at 18 months of age. Both the phosphorylated and unphosphorylated forms of the three proteins were increased, although the p-AMPK/AMPK, p-ULK1/ULK, and p-ACC/ACC ratios were similar in the untreated and treated animals (Appendix A), suggesting that AMPK was not a direct target of β-RA. Moreover, the 3T3-L1 cells treated with β-RA did not experience changes in the p-AMPK/AMPK ratio, with p-AMPK being almost undetectable in both the treated and untreated cells (Appendix A).

## 4. Discussion

β-RA is an HBA that shows powerful therapeutic benefits in CoQ deficiency mouse models caused by mutations in *Coq6*, *Coq7*, *Coq8b*, or *Coq9* [21,22,24,25]. Those studies administered high doses of oral β-RA, but the mechanisms have not been clearly elucidated in podocyte-specific *Coq6* or *Coq8b* knockout mice [24,25]. Moreover, chronic β-RA supplementation maintains a lower body weight in wild-type mice than untreated mice [21], but the causes and mechanisms of this effect were completely unknown. In our current work, we demonstrated that the therapeutic mechanism of β-RA in *Coq9^R239X^* mice was based on the capability of this molecule to enter the CoQ biosynthetic pathway and compete with 4-HB, resulting in a reduction of the levels of DMQ, an intermediate metabolite that is detrimental for mitochondrial function [44]. Moreover, our study revealed that β-RA prevented the accumulation of WAT during animal development and aging, thus preventing age-related hepatic steatosis. This powerful effect was due to an inhibition of preadipocyte proliferation and fat production, as well as the stimulation of lipolysis, gluconeogenesis, and glucose and acetyl-CoA utilization, mainly in the kidneys.

The fundamental rationale for the treatment with β-RA in primary CoQ deficiency is the induction of a bypass effect since β-RA has the hydroxyl group that is normally incorporated into the benzoquinone ring by the hydroxylase COQ7. Because COQ9 is essential for the stability and function of COQ7 [6], defects in either *Coq7* or *Coq9* are susceptible to be effectively treated by β-RA [1,21,22,23,45]. Surprisingly, β-RA treatment was also successful in podocyte-specific *Coq6* or *Coq8b* knockout mice, yet the mechanisms in those cases were apparently not related to a bypass effect, suggesting that the β-RA may induce additional therapeutic mechanisms. However, our results confirmed that the therapeutic mechanism of β-RA in the *Coq9^R239X^* mice was due to its action in CoQ metabolism, as demonstrated by (1) the decrease in the levels of DMQ, with the effect being more intense in the kidneys (the tissue that accumulated more β-RA), and (2) the suppression of the therapeutic effect of β-RA due to the co-administration of 4-HB, which attenuated the decrease of DMQ_9_, thus supporting the theory of competition between the molecules when trying to enter the CoQ biosynthetic pathway in vivo [38]. The results obtained with the co-administration of 4HB and β-RA also suggest that the K_M_ for β-RA was higher than the K_M_ for 4-HB in the prenylation reaction catalyzed by COQ2 [22,38]. Moreover, the therapeutic effects observed in this study were achieved with a third of the dose that was previously used [22]. Thus, the effects in this study were also similar to the results published in the *Coq7* conditional KO mice [23] despite the phenotypes of both models being substantially different [6,21]. This is important because animal studies that use lower doses of a drug could potentially be translatable to the human situation, decreasing the cost of the treatment and being more feasible regarding its administration, especially in the pediatric population. However, our results in the *Coq9^R239X^* mice showed that β-RA had limitations regarding inducing an increase in the levels of CoQ, suggesting that the co-supplementation of β-RA and CoQ_10_ could result in improved therapeutic outcomes [46]. Moreover, β-RA is not able to be modified the DMQ/CoQ ratio in the brain, suggesting that β-RA may have additional mechanisms that reduce the astrogliosis or that the effects on CoQ metabolism are happening in specific cells types or areas in the brain.

In wild-type animals, chronic β-RA supplementation prevented the accumulation of WAT. The in vitro experiments in this study demonstrated that β-RA inhibited preadipocytes proliferation, which is a result that was also achieved by other phenolic acids [47,48], including *p*-coumaric [47], which was reported to serve as a benzoquinone precursor for CoQ biosynthesis in humans and mice [49]. Whether the alteration on CoQ biosynthesis that was induced by β-RA, i.e., the decrease in CoQ levels or the mild accumulation of DMQ, may contribute to the accumulation of WAT remains to be elucidated. The anti-proliferative effect of β-RA in preadipocytes induces the downregulation of PPARγ, which seems to be critical for the suppression of adipocyte differentiation and the development of mature adipocytes [50]. Consequently, β-RA may act by preventing WAT hyperplasia and hypertrophy, both of which contribute to avoiding overweight and obesity in children and adults [51,52,53].

In addition to the direct effects of β-RA in adipocytes, in vivo experiments utilizing hypotheses that were generated by proteomic profiling, and following these observations up with focused validation experiments, showed a tissue metabolic switch, mainly in the kidneys. This tissue could account for up to 40% of the overall gluconeogenesis of the body under certain conditions, e.g., the post-absorptive phase [54,55], during which glycerol is one of the gluconeogenic renal precursors [54]. Although renal gluconeogenesis mainly serves to produce glucose only for its own utilization in the kidneys, this metabolic process can also participate in the regulation of systemic glucose metabolism [55]. Therefore, our results suggest that the β-RA induces renal gluconeogenesis from glycerol, and the resulting glucose is used in glycolysis to produce pyruvate and then acetyl-CoA, which is ultimately funneled into the TCA cycle. Acetyl-CoA may not only be produced through the classical pathway but also through an alternative pathway that involves α-ketoglutarate dehydrogenase and aldehyde dehydrogenase and uses acetaldehyde as an intermediate metabolite [56]. Interestingly, the production and use of acetyl-CoA in mitochondria were postulated as a metabolic signal of survival in organisms [57], which is consistent with a reduction in the WAT content [57,58], the stimulation of ketogenesis [57,59], the limitation of fatty acid synthesis, and the prevention of hepatic steatosis [57,58,59]. Nevertheless, it is unclear whether the metabolic effects in the kidneys and, to a lesser extent, in the liver are due to β-RA itself or whether they are the consequences of having a low amount of WAT. This second option could explain the downregulation of fatty acid β-oxidation in the kidneys and the subsequent preference for glucose metabolism. A potential regulator for all these metabolic changes is GSK3β, which is highly increased in the mitochondria of the treated wild-type animals. GSK3β regulates a variety of cellular processes, including glucose metabolism. In fact, its upregulation was associated with an amelioration of diabetes-induced kidney injury [60]. Consequently, these metabolic adaptations in the kidneys in response to chronic supplementation of β-RA could explain, at least in part, the positive therapeutic outcomes achieved in the podocyte-specific *Coq6* or *Coq8b* knockout mice [24,25] and open the potential application of β-RA in treating other renal metabolic diseases.

To conclude, the results reported here demonstrate that chronic supplementation with β-RA in mice induces different metabolic effects with relevant therapeutic implications for the treatment of primary CoQ deficiency and the prevention of age-related overweight and associated hepatic steatosis. The first application is based on the ability of β-RA to enter the CoQ biosynthetic pathway, compete with a lower affinity with the natural substrate 4-HB, and, consequently, reduce the levels of DMQ in cases of defects in *Coq9* or *Coq7*. The second application is based on a combination of direct influences over WAT, ultimately preventing the hyperplasia and hypertrophy of adipocytes, and to indirect systemic mechanisms, mainly by the adaptations of renal metabolism. Nevertheless, this study has some limitations: (1) although β-RA can prevent the accumulation of WAT during aging, it is unknown whether it can reduce WAT in already obese animals; (2) although this long-term study showed convincing therapeutic actions of β-RA, the effects of β-RA administration should be evaluated in mice with different genetic backgrounds and models of both diet-induced obesity and genetic-induced obesity; and (3) a minimal effective dose and potential dose-dependent specific effects must be defined for both therapeutic applications. Nevertheless, the data gathered in the present work are relevant for the future translation of the treatment with β-RA into the clinic, especially considering that we have shown the effects of the long-term administration of β-RA in a mouse model of age-related overweight and mitochondrial encephalopathy due to CoQ deficiency.

## Figures and Tables

**Figure 1 biomedicines-09-01457-f001:**
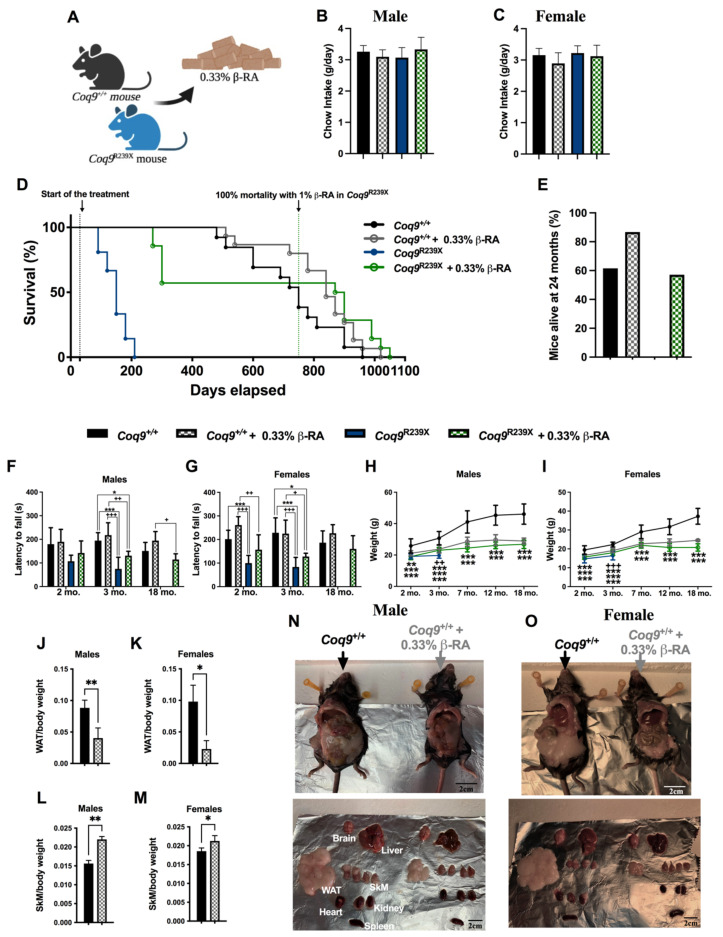
Survival and phenotypic characterization of *Coq9^+/+^* and *Coq9^R239X^* mice during the supplementation with 0.33% β-RA. (**A**) Schematic figure of the β-RA treatment in *Coq9*^+/+^ and *Coq9*^R239X^ mice. (**B**,**C**) Daily food intake in male and female *Coq9*^+/+^ and *Coq9*^R239X^ mice. (**D**) Survival curve of the *Coq9^+/+^* mice, *Coq9^+/+^* mice given 0.33% β-RA supplementation, *Coq9^R239X^* mice, and *Coq9^R239X^* mice given 0.33% β-RA supplementation. The treatments started at 1 month of age (log-rank (Mantel–Cox) test or Gehan–Breslow–Wilcoxon test; *Coq9*^+/+^ mice, *n* = 13; *Coq9^+/+^* mice under 0.33% β-RA supplementation, *n* = 15; *Coq9^R239X^* mice, *n* = 21; *Coq9^R239X^* mice under 0.33% β-RA supplementation, *n* = 14). (**E**) Percentage of mice alive at 24 months of age. (**F**,**G**) Rotarod test of male and female *Coq9^+/+^* mice, *Coq9^+/+^* mice given 0.33% β-RA supplementation, *Coq9^R239X^* mice, and *Coq9^R239X^* mice given 0.33% β-RA supplementation. (**H**,**I**) Body weight of male and female *Coq9^+/+^* mice, *Coq9^+/+^* mice given 0.33% β-RA supplementation, *Coq9^R239X^* mice, and *Coq9^R239X^* mice given 0.33% β-RA supplementation. (**J**–**M**) Weight of the epididymal, mesenteric, and inguinal white adipose tissue (WAT) (**J**,**K**) and hind legs skeletal muscle (SKM) (**L**,**M**) relative to the total body weight in male and female *Coq9^+/+^* mice, and *Coq9^+/+^* mice given 0.33% β-RA supplementation at 18 months of age. (**N**,**O**) Representative images of male (N) and female (O) mice and their tissues at 18 months of age, both untreated and treated. Data are expressed as mean ± SD. * *p* < 0.05, ** *p* < 0.01, *** *p* < 0.001, differences versus *Coq9^+/+^*; + *p* < 0.05, ++ *p* < 0.01, +++ *p* < 0.001, *Coq9^+/+^* mice given 0.33% β-RA supplementation (one-way ANOVA with Tukey’s post hoc test or Mann–Whitney (nonparametric) test; *n* = 5–34 for each group).

**Figure 2 biomedicines-09-01457-f002:**
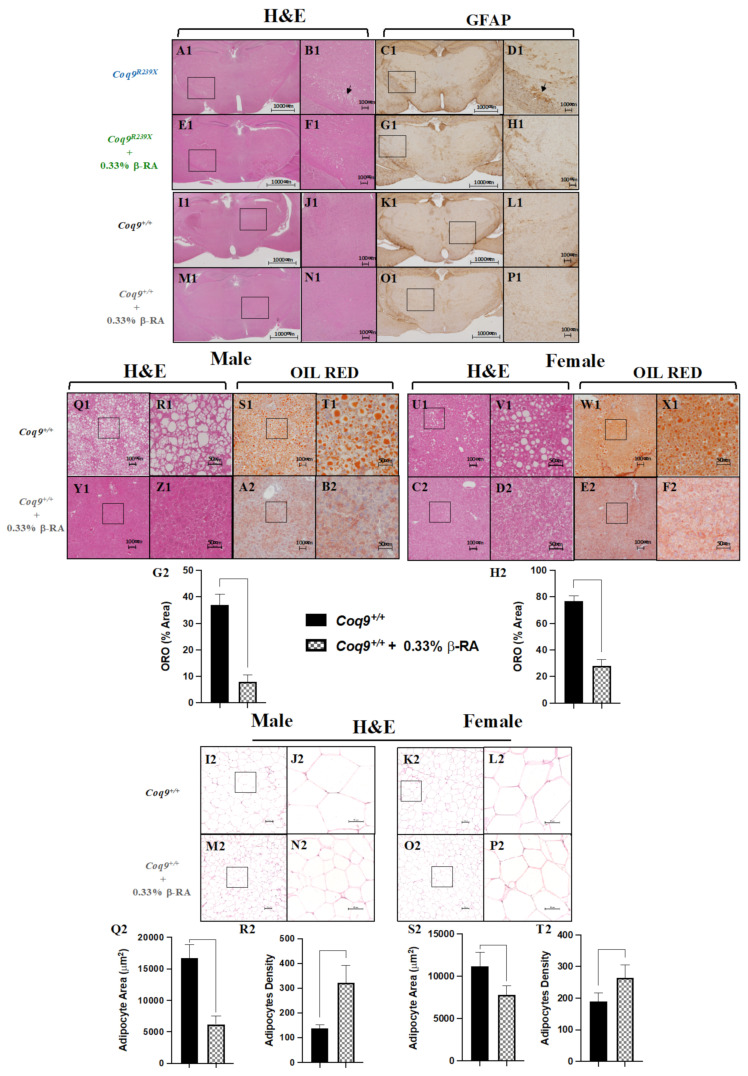
Morphological evaluation of symptomatic tissues from *Coq9^R239X^* and *Coq9^+/+^* mice under the supplementation with 0.33% β-RA. (**A1**–**P1**) H&E stain and anti-GFAP immunohistochemistry in sections of the diencephalon from *Coq9^R239X^* mice (**A1**–**D1**), *Coq9^R239X^* mice given 0.33% β-RA supplementation (**E1**–**H1**), *Coq9^+/+^* mice (**I1**–**L1**), *Coq9^+/+^* mice given 0.33% β-RA supplementation (**M1**–**P1**) at 3 months of age. Scale bars: 1000 μm left, 100 μm right. Black arrows show areas of spongiosis and astrogliosis. (**Q1**–**F2**) H&E and Oil Red stains in sections of the liver at 18 months of age from male (**Q1**–**T1**) and female (**U1**–**X1**) *Coq9^+/+^* mice and male (**Y1**–**B2**) and female (**C2**–**F2**) *Coq9^+/+^* mice given 0.33% β-RA supplementation. Scale bars: 100 μm left, 50 μm right. (**G2**–**H2**) Percentage of the area corresponding to the Oil Red O stains in sections of the liver at 18 months of age from *Coq9^+/+^* mice and *Coq9^+/+^* mice given 0.33% β-RA supplementation. (**I2**–**P2**) H&E stains in sections of the epididymal WAT at 18 months of age from male (**G2,H2**) and female (**I2,J2**) *Coq9^+/+^* mice and male (**K2,L2**) and female (**M2**,**N2**) *Coq9^+/+^* mice given 0.33% β-RA supplementation. Scale bars: 100 μm left, 50 μm right. (**Q2–T2**) Average of the area of each adipocyte and the adipocytes density in sections of the epididymal WAT at 18 months of age from *Coq9^+/+^* mice and *Coq9^+/+^* mice given 0.33% β-RA supplementation. Data are expressed as mean ± SD. **p* < 0.05, differences versus *Coq9^+/+^* (Mann–Whitney (nonparametric) test; *n* = 4–6 for each group).

**Figure 3 biomedicines-09-01457-f003:**
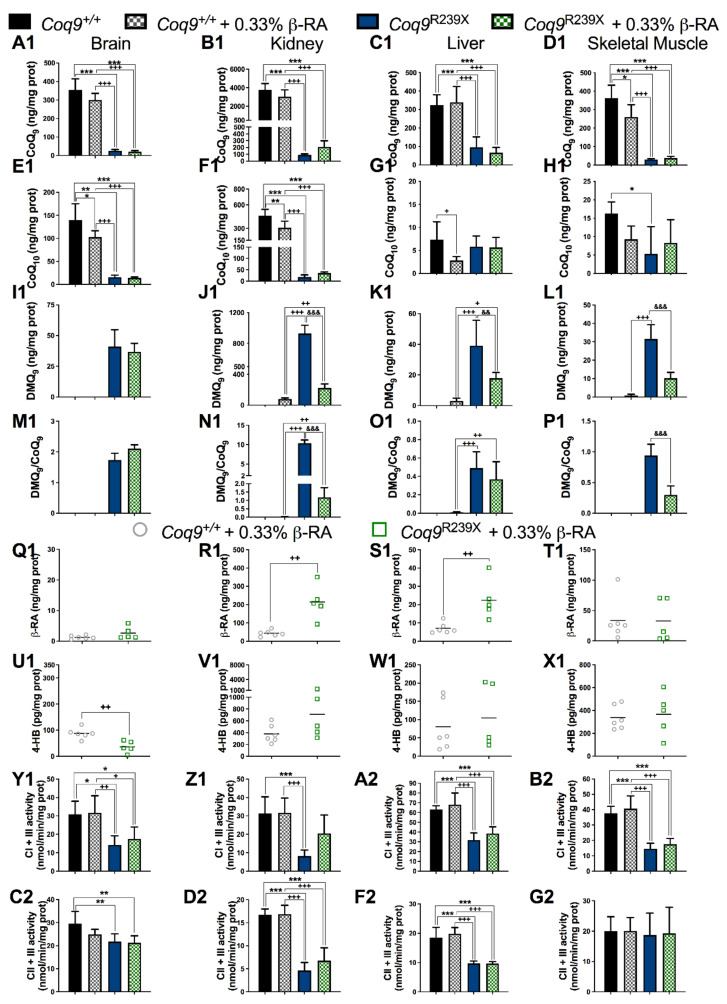
CoQ metabolism and mitochondrial function in the tissues from the *Coq9^+/+^* mice, *Coq9^+/+^* mice given supplementation with 0.33% β-RA, *Coq9^R239X^* mice, and *Coq9^R239X^* mice given supplementation with 0.33% β-RA. (**A1**–**D1**) Levels of CoQ_9_ in the brain (**A1**), kidneys (**B1**), liver (**C1**), and hind legs skeletal muscle (**D1**) from the *Coq9^+/+^* mice, *Coq9^+/+^* mice given the 0.33% β-RA treatment, *Coq9^R239X^* mice, and *Coq9^R239X^* mice given the 0.33% β-RA treatment. (**E1**–**H1**) Levels of CoQ_10_ in the brain (**E1**), kidneys (**F1**), liver (**G1**) and hind legs skeletal muscle (**H1**) from the *Coq9^+/+^* mice, *Coq9^+/+^* mice given the 0.33% β-RA treatment, *Coq9^R239X^* mice, and *Coq9^R239X^* mice given the 0.33% β-RA treatment. (**I1**–**L1**) Levels of DMQ_9_ in the brain (**I1**), kidneys (**J1**), liver (**K1**), and hind legs skeletal muscle (**L1**) from *Coq9^+/+^* mice, *Coq9^+/+^* mice given the 0.33% β-RA treatment, *Coq9^R239X^* mice, and *Coq9^R239X^* mice given the 0.33% β-RA treatment. Note that DMQ_9_ was not detected in samples from the *Coq9^+/+^* mice. (**M1**–**P1**) DMQ_9_/CoQ_9_ ratio in the brain (**M1**), kidneys (**N1**), liver (**O1**), and hind legs skeletal muscle (**P1**) from the *Coq9^+/+^* mice, *Coq9^+/+^* mice given the 0.33% β-RA treatment, *Coq9^R239X^* mice, and *Coq9^R239X^* mice given the 0.33% β-RA treatment. (**Q1**–**X1**) Levels of β-RA in the brain (**Q1**), kidneys (**R1**), liver (**S1**), and hind legs skeletal muscle (**T1**) from the *Coq9^+/+^* mice given the 0.33% β-RA treatment and *Coq9^R239X^* mice given the 0.33% β-RA treatment. β-RA was undetectable in the *Coq9^+/+^* mice and *Coq9^R239X^* mice. (**U1**–**X1**) Levels of 4-HB in the brain (**U1**), kidneys (**V1**), liver (**W1**), and hind legs skeletal muscle (X1) from the *Coq9^+/+^* mice given the 0.33% β-RA treatment and *Coq9^R239X^* mice given the 0.33% β-RA treatment. (**Y1**–**B2**) Complex I + III (CI + III) activities in the brain (**Y1**), kidneys (**Z1**), liver (**A2**), and hind legs skeletal muscle (**B2**) from the *Coq9^+/+^* mice, *Coq9^+/+^* mice given the 0.33% β-RA treatment, *Coq9^R239X^* mice, and *Coq9^R239X^* mice given the 0.33% β-RA treatment. (**C2**–**G2**) Complex II + III (CII + III) activities in the brain (**C2**), kidneys (**D2**), liver (**F2**), and hind legs skeletal muscle (**G2**) from the *Coq9^+/+^* mice, *Coq9^+/+^* mice given the 0.33% β-RA treatment, *Coq9^R239X^* mice, and *Coq9^R239X^* mice given the 0.33% β-RA treatment. Tissues from the mice at 3 months of age. Data are expressed as mean ± SD. * *p* < 0.05, ** *p* < 0.01, *** *p* < 0.001, differences versus *Coq9^+/+^*. + *p* < 0.05, ++ *p* < 0.01, +++ *p* < 0.001, differences versus the *Coq9^+/+^* mice given the 0.33% β-RA treatment. && *p* < 0.01, &&& *p* < 0.001, differences versus *Coq9^R239X^*. One-way ANOVA with Tukey’s post hoc test or Mann–Whitney (nonparametric) test; *n* = 5–8 for each group.

**Figure 4 biomedicines-09-01457-f004:**
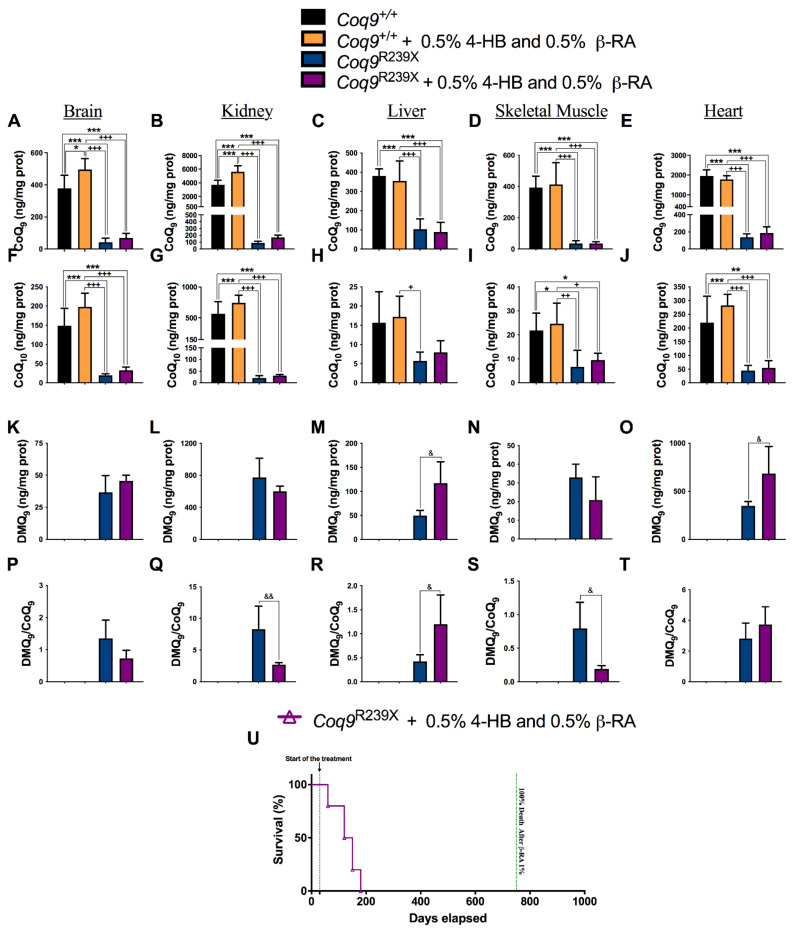
Co-administration of 4-HB suppressed the effects of the β-RA treatment in the *Coq9^+/+^* and *Coq9^R239X^* mice. (**A**–**E**) Levels of CoQ_9_ in the brain (**A**), kidneys (**B**), liver (**C**), skeletal muscle (**D**), and heart (**E**) from the *Coq9^+/+^* mice, *Coq9^+/+^* mice given the 0.5% 4-HB + 0.5% β-RA treatment, *Coq9^R239X^* mice, and *Coq9^R239X^* mice given the 0.5% 4-HB + 0.5% β-RA treatment. (**F**–**J**) Levels of CoQ_10_ in the brain (**F**), kidneys (**G**), liver (**H**), skeletal muscle (**I**), and heart (**J**) from the *Coq9^+/+^* mice, *Coq9^+/+^* mice given the 0.5% 4-HB + 0.5% β-RA treatment, *Coq9^R239X^* mice, and *Coq9^R239X^* mice given the 0.5% 4-HB + 0.5% β-RA treatment. (**K**–**O**) Levels of DMQ_9_ in the brain (**K**), kidneys (**L**), liver (**M**), skeletal muscle (**N**), and heart (**O**) from the *Coq9^+/+^* mice, *Coq9^+/+^* mice given the 0.5% 4-HB + 0.5% β-RA treatment, *Coq9^R239X^* mice, and *Coq9^R239X^* mice given the 0.5% 4-HB + 0.5% β-RA treatment. (**P**–**T**) The DMQ_9_/CoQ_9_ ratio in the brain (**P**), kidneys (**Q**), liver (**R**), skeletal muscle (**S**), and heart (**T**) from the *Coq9^+/+^* mice, *Coq9^+/+^* mice given the 0.5% 4-HB + 0.5% β-RA treatment, *Coq9^R239X^* mice, and *Coq9^R239X^* mice given the 0.5% 4-HB + 0.5% β-RA treatment. (**U**) Survival curve of the *Coq9^R239X^* mice given the 0.5% 4-HB + 0.5% β-RA treatment. Tissues from mice at 3 months of age. Data are expressed as mean ± SD. * *p* < 0.05, ** *p* < 0.01, *** *p* < 0.001, differences versus *Coq9^+/+^*. + *p* < 0.05, ++ *p* < 0.01, +++ *p* < 0.001, differences versus *Coq9^+/+^* after the 0.5% 4-HB and 0.5% β-RA treatment. & *p* < 0.05, && *p* < 0.01, differences versus *Coq9^R239X^*. One-way ANOVA with Tukey’s post hoc test or Mann–Whitney (nonparametric) test; *n* = 5–10 for each group.

**Figure 5 biomedicines-09-01457-f005:**
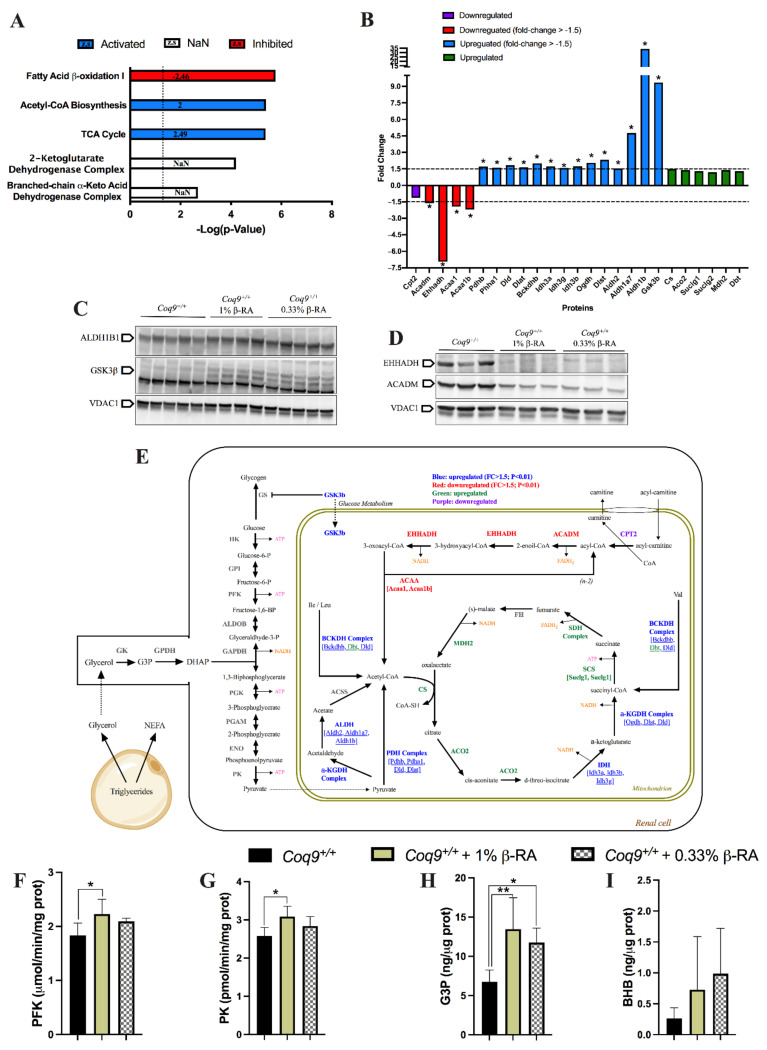
Adaptation of the mitochondrial proteome to the treatment with β-RA in the kidneys of the *Coq9^+/+^* mice. (**A**) Top enriched metabolic canonical pathways in the renal mitochondrial proteome from the *Coq9^+/+^* mice after two months of 1% β-RA supplementation. Dotted line: Adjusted *p* = 0.05. Blue signifies that the category was expected to be activated according to the z-score; red signifies that the category was expected to be inhibited according to the z-score. (**B**) Fold change (treated/untreated) of the proteins involved in the identified enriched metabolic canonical pathways in the renal mitochondrial proteome. Purple signifies proteins that were downregulated; red signifies proteins that were downregulated with a fold change > 1.5; blue signifies proteins that were upregulated with a fold change > 1.5; green signifies proteins that were upregulated. * *p* < 0.05. Mitochondrial proteomics was performed in isolated mitochondria. (**C**,**D**) Western blot of some key proteins identified in the proteomics analysis to validate the changes observed with the treatment of β-RA. The validation was performed with the treatment of β-RA at 1% and extended to the treatment of β-RA at 0.33%. The selected proteins were ALDH1B1, GS3Kβ, EHHADH, and ACADM. VDAC1 was used as a loading control. The experiments were performed in tissue homogenate. (**E**) Schematic figure of the most important changes in the mitochondrial proteomes from the kidneys of the *Coq9^+/+^* mice after the β-RA treatment. (**F**,**G**) Activities of the glycolytic enzymes phosphofructokinase (PFK) (**F**) and pyruvate kinase (PK) (**G**) in the kidneys of the *Coq9^+/+^* mice treated with β-RA at 1 and 0.33%. (**H**,**I**) Levels of glycerol-3-phosphate (G3P) (**H**) and β-hydroxybutyrate (BHP) (**I**) in the kidneys of the *Coq9^+/+^* mice treated with β-RA at 1 and 0.33%. Tissues from mice at 3 months of age. Data are expressed as mean ± SD. * *p* < 0.05, ** *p* < 0.01, differences versus *Coq9^+/+^*. One-way ANOVA with Tukey’s post hoc test or Mann–Whitney (nonparametric) test; *n* = 5–7 for each group.

**Figure 6 biomedicines-09-01457-f006:**
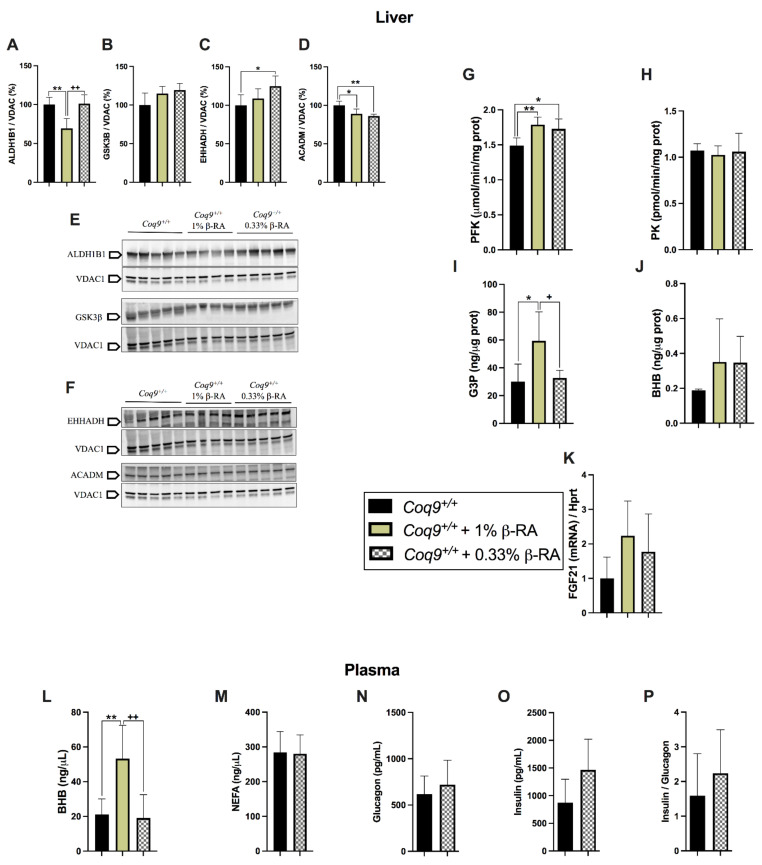
Metabolic characterization of liver and plasma after the treatment with β-RA in the *Coq9^+/+^* mice. (**A**–**F**) Levels of the proteins ALDH1B1 (**A**,**E**), GSK3β (**B**,**E**), EHHADH (**C**,**F**), and ACADM (**D**,**F**) in the liver of the *Coq9^+/+^* mice treated with β-RA at 1 and 0.33%. VDAC1 was used as a loading control. The experiments were performed in tissue homogenate. (**G**–**J**) Activities of the glycolytic enzymes phosphofructokinase (PFK) (**G**) and pyruvate kinase (PK) (**H**) in the liver; levels of glycerol-3-phosphate (G3P) in the liver (**I**); levels of β-hydroxybutyrate (BHP) in the liver (**J**). (**K**) Levels of the FGF21 mRNA in the liver. (**L**–**P**) Levels of β-hydroxybutyrate (BHP) (**L**), non-esterified fatty acids (NEFA) (**M**), glucagon (**N**), and insulin (**O**) in the plasma of the *Coq9^+/+^* mice treated with β-RA; glucagon/insulin ratio (**P**) in the plasma of the *Coq9^+/+^* mice treated with β-RA. Tissues from mice at 3 months of age. Data are expressed as mean ± SD. * *p* < 0.05, ** *p* < 0.01, differences versus *Coq9^+/+^*. + *p* < 0.05, ++ *p* < 0.01, differences versus *Coq9^+/+^* after 1% β-RA treatment. One-way ANOVA with Tukey’s post hoc test or Mann–Whitney (nonparametric) test; *n* = 5–7 for each group.

**Figure 7 biomedicines-09-01457-f007:**
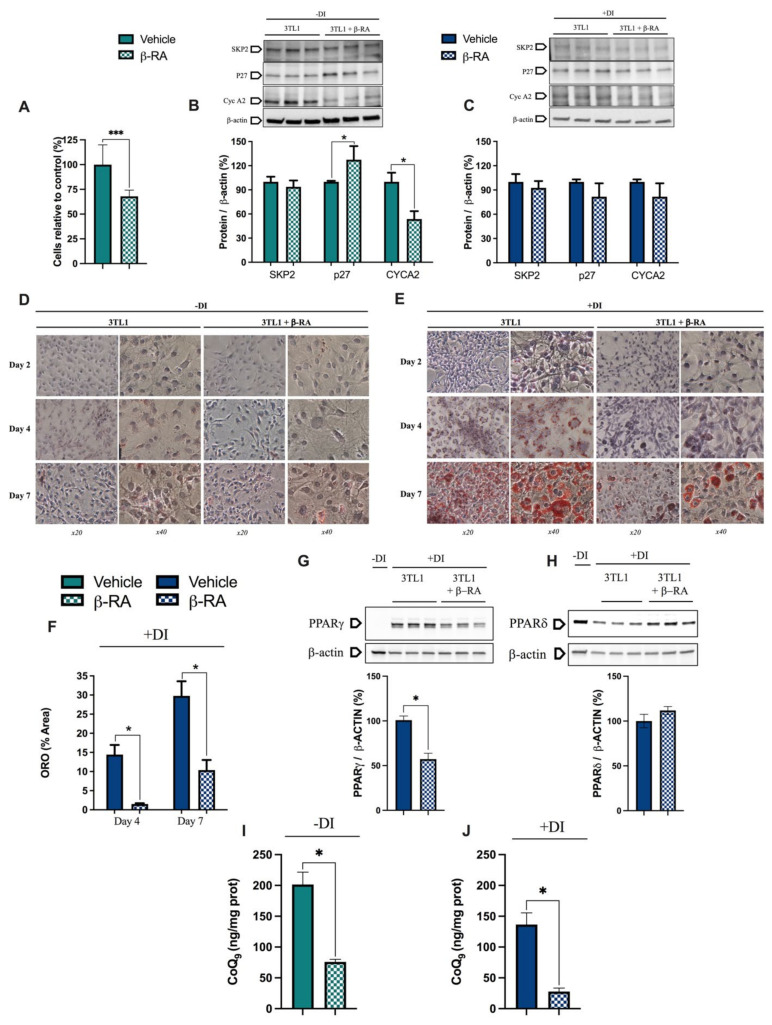
Direct effects of β-RA on adipogenesis. (**A**) Percentage of 3TL1 cells after seven days of treatment with 1 mM β-RA relative to the number of untreated 3TL1 cells. Cells cultured in proliferative conditions. (**B**) Levels of the proteins SKP2, p27, and CYCA2, which were involved in the control of the cell cycle. The 3TL1 cells were treated for seven days with 1 mM β-RA in proliferative conditions. (**C**) Levels of the proteins SKP2, p27, and CYCA2, which were involved in cell cycle control. The 3TL1 cells were treated for seven days with 1 mM β-RA in differentiative conditions. (**D**,**E**) Oil Red O staining in 3TL1 cells cultured under proliferative (**D**) and proliferative + differentiative (**F**) conditions. The 3TL1 cells were treated with 1 mM β-RA from day 0 in both conditions and the stains were performed on three different days (2, 4, and 7). (**F**) Percentage of the area corresponding to the Oil Red O stains in the 3TL1 cells in differentiative conditions after days 4 and 7 of treatment with 1 mM β-RA. (**G**,**H**) Levels of PPARγ and PPARδ in the 3TL1 cells cultured in proliferative + differentiative (**F**) conditions and treated with 1 mM β-RA. The results in non-differentiated cells are shown in line one as the negative control. (**I**,**J**) Levels of CoQ_9_ in the 3TL1 cells cultured in proliferative conditions (**I**) and differentiative conditions (**J**) and treated with 1 mM β-RA. Data are expressed as mean ± SD. * *p* < 0.05, *** *p* < 0.001, differences versus untreated cells (Mann–Whitney (nonparametric) test; *n* = 6 for each group).

## Data Availability

The mass spectrometry proteomics data were deposited to the ProteomeXchange (http://www.proteomexchange.org/ accessed on 1 April 2020). Consortium via the PRIDE partner repository with the dataset identifier PXD018311 (1 April 2020).

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
