# Peer review of "β-RA Targets Mitochondrial Metabolism and Adipogenesis, Leading to Therapeutic Benefits against CoQ Deficiency and Age-Related Overweight"

_biomedicines, 2021, doi:10.3390/biomedicines9101457_

Round 1
Reviewer 1 Report
In the article entitled, ‘b-RA targets mitochondrial metabolism and adipogenesis, leading to therapeutic benefits against CoQ deficiency and age-related overweight’, Hidalgo-Gutiérrez et al., have assessed the effect of b-RA in the Coq+/+ and Coq9R239X mice model of primary mitochondrial encephalopathy and on adipogenesis. The authors have analyzed the effects of different treatments of b-RA on mitochondrial metabolism and adipogenesis using biomarkers and histopathology experiments along with an exhaustive proteomic analysis. The authors demonstrate that b-RA treatment results in increased CoQ10 by bypassing the Coq7 step and competing with 4-HB to enter the CoQ biosynthetic pathway resulting in a reduction of the levels of DMQ that is detrimental for mitochondrial function. Using proteomic study, they also show that alterations in various metabolic pathways could be the underlying cause of the observed changes. They have further shown that b-RA treatment results in a reduction in WAT which might have implications in the treatment of obesity. The numerous effects of b-RA on metabolism regulating mitochondrial metabolism as well as adipogenesis are very interesting and hold significance in the treatment of metabolic disorders associated with mitochondrial dysfunction and obesity. The authors have studied the effects of b-RA in multiple tissues. But, the biological significance of the findings in multiple tissues is not forthcoming and the differences in results between different tissues have not been discussed/justified appropriately. In many instances, the flow of results is very inconsistent and the conclusions are not very clear. However, the study is interesting and the authors have utilized appropriate methodologies. There are a few concerns that need to be addressed before the article is accepted for publication.
Major concerns:
- The authors must include the effects of b-RA supplementation on morphological alterations in the brain, liver, kidneys, spleen, heart, or small intestine, and the blood and urine markers of renal and hepatic functions at 18 months in the Coq9+/+ and Coq9R239X
- What is the status of steatosis in the Coq+/+ mice model? The authors must include the Oil Red staining data for the same.
- The authors have observed differences upon morphological evaluation of the encephalon from Coq9R239X and Coq9+/+ mice under the supplementation with 0.33% b-RA in Figure 2 (A1-H1). However, in figure 3 the authors do not see a significant difference in the levels of Coq10 and DMQ9 between Coq9R239X and Coq9+/+ Can the authors justify why the effects are more prominent morphologically and not upon quantitation of absolute levels of the Coqs?
- The result section for figures 3 and 4 are not written very coherently and the significance of the results is not clear in the different tissues. The authors must re-write these results emphasizing the findings and their implications.
- Figure 4 U: The authors must include the survival curve for 0.5% b-RA treated mice alone in the Kaplan Meier plot and provide statistical analysis.
- While the regulation of enzyme expression seems promising, it does not ensure an increase (or decrease) in the activity of the enzyme. Did the authors perform metabolomics analyses to confirm the findings of the quantitative proteomics?
- The authors have utilized different doses of b-RA in different experiments and different treatment durations. Can the authors briefly describe the effects of dose and treatment kinetics of b-RA on mitochondrial metabolism and adipogenesis?
- What is the status of WAT in the Coq9R239X mice?
- The transition from Coq10 biosynthesis to WAT synthesis seems a little abrupt and correlations can not be drawn.
- While the effect of b-RA on WAT levels is very drastic and interesting, the mechanism for the b-RA-mediated reduction in WAT is not evident from the results. The authors must comment on the mechanism.
- Figure 7H: The increase in PPARd is not evident and a better representative blot must be included.
Minor concerns:
- The authors must include details about the Coq9R239X mice model in the introduction section.
- Figure 1N and O: The authors must include a scale to compare the size of the tissues extracted between Coq9+/+ and Coq+/+ + 0.33% b-RA. The authors must also label the tissues.
- The significance of staining with GFAP must be included in the text for Figure 2.
- Scale bar must be mentioned in the legend for Figure 2.
- In figure 3, the color of the bars is not very clear. The authors must label the x-axis properly.
- The representation of statistical significance in the different figures is not clear. The authors could use lines to show the comparisons between different bars for statistical significance.
- The resolution of Figure 5 is very poor and a better-resolved image must be included. The words are not legible.
Reviewer 2 Report
The manuscript “beta-RA targets mitochondrial metabolism and adipogenesis, leading to therapeutic benefits against CoQ deficiency and age-related overweight” by Hidalgo-Gutiérrez et al presents the results of mouse studies aimed at determining the effects of natural phenolic compound to improves mitochondrial dysfunction. The manuscript is well written. The authors used a wide range of physiological and biochemical methods, which adds to the significance of the article. Despite the high quality of the work, questions remain about experimental procedures, statistical processing and writing of the manuscript.
- It is not entirely clear how the authors produced the isolation of mitochondria for the further measurement of the respiration rate. Was the protocol common to both the kidneys and the brain? Obtaining intact brain mitochondria is a laborious process and requires a Percoll gradient centrifugation step. Ignoring this step will lead to mitochondrial contamination with synaptosomes and myelin. See Sims and Anderson (2008) PMID: 18600228. Isolation of mitochondria from the brain also possible using digitonin (Rosenthal et al., 1987; PMID: 3693430). The authors should describe in more detail the method of isolating mitochondria from the brain and kidneys. Unfortunately, more detailed information on mitochondrial isolation is also lacking in the cited references.
- The authors do not give the value of the respiratory control ratio (RCR) of mitochondria (state3 / state 4). RCR is the simplest and most effective way to assess the quality of mitochondrial preparation.
- There is no question why the authors used ANOVA with a Tukey’s post hoc test for compare the differences between three experimental groups. Why did the authors use the Student's t-test when comparing the two experimental groups? Authors used five-ten experiments per group, which is a small sample. Authors should prove that the distribution in their groups did not differ from normal (Shapiro-Wilk’s W test or Kolmogorov-Smirnov test), or use nonparametric statistical significance tests, for example, the Mann-Whitney test.
- The full name should be added to some abbreviations (FCCP, EGTA, EDTA, TCA, etc.)
- Manufacturers of some reagents are not indicated (Beta-Resorcylic acid, FCCP, ADP, HEPES, EDTA, EGTA, Antimycin, Oligomycin, etc.).
- Figure 5 is in low resolution, difficult to read.
Round 2
Reviewer 1 Report
The authors have addressed the concerns raised. The manuscript may be accepted in the present form.